# Allelopathic Potential of Mangroves from the Red River Estuary against the Rice Weed *Echinochloa crus-galli* and Variation in Their Leaf Metabolome

**DOI:** 10.3390/plants11192464

**Published:** 2022-09-21

**Authors:** Dounia Dhaou, Virginie Baldy, Dao Van Tan, Jean-Rémi Malachin, Nicolas Pouchard, Anaïs Roux, Sylvie Dupouyet, Stéphane Greff, Gérald Culioli, Thomas Michel, Catherine Fernandez, Anne Bousquet-Mélou

**Affiliations:** 1IMBE, Aix Marseille University, Avignon University, CNRS, IRD, 13331 Marseille, France; 2Department of Genetics-Biochemistry, Faculty of Biology, Hanoi National University of Education (HNUE), 131000 Hanoi, Vietnam; 3Institut de Chimie de Nice, Université Côte d’Azur, CNRS, UMR 7272, 06108 Nice, France

**Keywords:** allelopathy, mangrove, barnyard grass (*Echinochloa crus-galli*), rice (*Oryza sativa*), metabolomics, UHPLC-ESI/qToF, *Aegiceras corniculatum*, *Sonneratia apetala*, sulfated ellagic acid, flavonoids, saponins

## Abstract

Mangroves are the only forests located at the sea–land interface in tropical and subtropical regions. They are key elements of tropical coastal ecosystems, providing numerous ecosystem services. Among them is the production of specialized metabolites by mangroves and their potential use in agriculture to limit weed growth in cultures. We explored the in vitro allelopathic potential of eight mangrove species’ aqueous leaf extracts (*Avicennia marina*, *Kandelia obovata*, *Bruguiera gymnorhiza*, *Sonneratia apetala*, *Sonneratia caseolaris*, *Aegiceras corniculatum*, *Lumnitzera racemosa* and *Rhizophora stylosa*) on the germination and growth of *Echinochloa crus-galli*, a weed species associated with rice, *Oryza sativa*. Leaf methanolic extracts of mangrove species were also studied via UHPLC-ESI/qToF to compare their metabolite fingerprints. Our results highlight that *A. corniculatum* and *S. apetala* negatively affected *E. crus-galli* development with a stimulating effect or no effect on *O. sativa*. Phytochemical investigations of *A. corniculatum* allowed us to putatively annotate three flavonoids and two saponins. For *S. apetala*, three flavonoids, a tannin and two unusual sulfated ellagic acid derivatives were found. Some of these compounds are described for the first time in these species. Overall, *A. corniculatum* and *S. apetala* leaves are proposed as promising natural alternatives against *E. crus-galli* and should be further assessed under field conditions.

## 1. Introduction

According to the United Nations Food and Agriculture Organization, about USD 95 billion losses in crop yields are caused by weeds annually worldwide [1]. Weeds exhibit several traits promoting their rapid development and adaptation to agronomic environment such as rapid growth rates, prolonged dormancy and ease of dispersal [2]. Manual weeding is the most effective means of control, but it is very costly in terms of labor. The ease of application of synthetic herbicides and the rapid effects on the target species have given rise to a rapid increase in their use, generating other problems in the management of agrosystems [3]. Indeed, such agricultural practices have led to environmental pollution, health hazards [4] and the development of resistance to these herbicides among weeds.

Rice is a staple food for billions of people around the world. One of the major challenges to global food security today is the growing world population, which requires an ever-increasing food supply [5]. In mainland Southeast Asia, rice cultivation provides self-sufficiency and food security for over 182 million people in Cambodia, Laos, Thailand and Vietnam [5]. Rice production is essential to Vietnam’s agricultural sector, accounting for 30% of the country’s total agricultural production value [6]. Since the 1990s, rice production has increased by 5.6% per year thanks to the increase in yields and in planted areas. The total area of rice fields is more than 7.7 million hectares, mainly located in the Mekong Delta and Red River Delta regions [6].

Among the genus *Echinochloa* (Poaceae) or barnyard grass, *Echinochloa crus-galli* (L.) P. Beauv. is the most prevalent weed species associated with rice cultivation [7]. The morphological similarity between *E. crus-galli* and rice (*Oryza sativa* L.) at the seedling stage leads to difficulty in recognition during manual weeding [8]. Moreover, *E. crus-galli* is one of the world’s most pernicious herbicide-resistant weeds [9]. Competition due to *E. crus-galli* causes about 35% of the worldwide losses in yield in rice [10]. For several years, intensified efforts have been made to identify environmentally benign chemical solutions for weed control, particularly through the use of promising allelopathic compounds produced by plants [11].

Plant allelopathy is a chemical interaction mediated by biochemicals released in the environment by a plant and that influences the growth and establishment of other plant species [12]. Allelopathic compounds are released into the environment through foliar leachates, root exudation, leaf litter decomposition or volatilization [12]. Neighboring plants can either be affected directly through various physiological processes (e.g., photosynthesis, nutrient uptake, cell division or elongation [13]) or by indirect effects such as the disruption of nitrogen mineralization and the inhibition of ectomycorrhizal fungi [14]). The main life stages typically affected by allelopathic compounds are seed germination and seedling growth [15]. Common negative allelopathic effects include seed germination inhibition [16], delayed germination [17] and seedling growth inhibition [18].

Plant allelopathy can have an important role in vegetation dynamics in both natural [19,20] and agricultural [21] ecosystems. Special attention has been paid to this process in crop ecosystems for several years as it can be an alternative weed management strategy in crop production with less environmental cost than the use of synthetized herbicides [22]. The most-explored way, to date, has been the selection of rice varieties that exhibit an allelopathic effect on barnyard grass [23,24,25].

Mangrove ecosystems cover 75% of the world’s subtropical and tropical coastlines, distributed between the latitudes 33 N and 38 S [26]. They occur only in intertidal zones, and have therefore developed the ability to grow under stressful conditions such as unusually high and variable salinity, inundation, soil anoxia and substrate movements [27]. They offer a wide range of services such as coastal erosion prevention, storm and flood reduction, water quality maintenance and support of wildlife. The use of natural compounds produced by trees in both agriculture and human health can be a part of mangrove ecosystem services [28]. Most natural compounds are specialized metabolites produced by plants to face stressful abiotic environment such as in mangrove ecosystems. These compounds can be potentially allelopathic after their release into the environment [29]. Little is known about the allelopathic potential of mangrove species: Chen & Peng [30] tested the allelopathic potential of three mangrove species on model target plants and another study suggested that allelopathic processes could be involved in mangrove succession [31]. To date, only a few studies have reported the allelopathic potential of mangroves species in weed management [32,33].

The recent development of plant metabolomics gives rise to new opportunities in describing a sample’s large set of metabolites and changes in their relative concentrations according to internal or external factors [34]. With an untargeted metabolomics approach, multivariate analysis of the resulting datasets allows researchers to rapidly compare the global metabolic contents of plants and to highlight the most significant biomarkers on which the annotation effort must be made. Ultra-high-performance liquid chromatography coupled with high-resolution mass spectrometry (UHPLC-HRMS) is a sensitive and powerful analytical tool that is now increasingly used for untargeted metabolomics [35,36]. Few studies have applied metabolomics for mangrove phytochemical description [37,38,39,40]. Such an analytical approach may provide valuable information for better description of plant natural products that might be involved in allelopathic activities.

The objectives of this study were (i) to compare the leaf chemical composition (focused on specialized metabolites) of eight mangrove species occurring in northern Vietnam based on an untargeted metabolomics study, (ii) to evaluate their allelopathic potential on the germination and growth of *E. crus-galli*, and (iii) to check that the potential effects observed on *E. crus-galli* are not toxic for *O. sativa*. We then looked for candidate plant species that can produce phytochemicals to be used as alternatives to synthetic herbicides in rice crops. In this case, these species should show negative effects on *E. crus-galli* and positive or neutral effects on *O. sativa*. Finally, biomarkers associated with these species of interest were putatively annotated.

## 2. Results

### 2.1. Comparison of Source Species Chemical Fingerprints

After mangrove leaf methanolic extract analysis using an untargeted UHPLC-MS-based metabolomics approach (the resulting chromatograms are shown in the Appendix A), data preprocessing allowed us to obtain four 616 *m*/*z* features which, after filtering, gave a final dataset accounting for 399 *m*/*z* features. The resulting PCA score plot (Figure 1a) revealed differences in the tested source species’ chemical fingerprints (PERMANOVA, *p* < 0.01). Four different groups were distinguished: (i) species belonging to the *Sonneratia* genus, separated from all the other species on the first component (36% of total variance); (ii) Species belonging to the Rhizophoraceae family, also separated from all other species on the first component, with *K. obovata* and *R. stylosa* being the only overlapped clusters in this representation; (iii) the group of *A. marina* and *A. corniculatum*, separated from all the other source species on the second component (12.4% of total variance); and finally, (iv) *L. racemosa*, observed at a central position.

The hierarchical clustering heatmap (Figure 1b) shows a similar grouping of samples, with *Sonneratia* species forming the most distant composition from all the other assessed species. The features were also clustered according to their quantitative similarities between groups. From top to bottom, the heatmap highlights 25 features found to be significantly more abundant in the *Sonneratia* genus. Among them, seven were also found to be more abundant in *L. racemosa* and three specifically in *S. apetala*. The following eight features were associated with *B. gymnorhiza*, followed by four features associated with *A. corniculatum*, two features for *S. caseolaris*, two features for *L. racemosa*, three features for *A. marina* and one feature for *R. stylosa.* Finally, species belonging to the Rhizophoraceae family shared three common features that were specifically more abundant (three bottom features).

### 2.2. Relative Allelopathic Effects

We chose to test the allelopathic potential of these eight mangrove species using aqueous leaf extracts. The stock solution was prepared by soaking 5 g (dry weight) in 100 mL of deionized water (5%) for 24 h in the dark. A dilute solution (2.5%) was prepared from the stock solution.

#### 2.2.1. Effects on Seed Germination

The germination velocity of *E. crus-galli* was significantly affected by the tested source species (F = 14.7, *p* < 0.05) without any effect of extract dose (F = 0.1, *p* = 0.74). For both doses (2.5 and 5%), the germination velocity was particularly decreased when plants were watered with *A. marina* extract (−13% for *E. crus-galli*). The highest germination velocity RAEs were obtained when *E. crus-galli* seeds were watered with *R. stylosa* extract, with a mean of −2%. For the *E. crus-galli* germination rates, the allelopathic effects ranged from −79% to 0% (Figure 2). The germination rates were significantly affected by both source species and dose (Table 1), with RAEs, on average, lower than −50% for *B. gymnorhiza* and *A. marina* (5% only) extracts, and between −20% and −50% for *S. apetala* and *A. corniculatum* (5% only) extracts (from lowest to highest).

As for *E. crus-galli*, the germination velocity of *O. sativa* was particularly decreased when the seeds were watered with *A. marina* extract (−8% for *O. sativa*). For the germination rates of *O. sativa*, the source species effect depended on the dose (Table 1), with seedling germination RAEs below −50% for *B. gymnorhiza* (2.5 and 5%) and below −20% for *A. marina* (5%) aqueous extracts only. For all the other tested source species and doses, the germination rate RAEs of *O. sativa* ranged from −11% to +3%, two of which were found to be positive for *A. corniculatum* (2.5%) and *L. racemosa* (5%) extracts.

#### 2.2.2. Effects on Seedling Growth

The aqueous extracts of the tested species generally affected seedling growth more than germination, but these effects depended on the target species, source species and extract dose. *E. crus-galli*’s hypocotyl length and biomass were significantly affected by the tested source species (F = 89.3, *p* < 0.01 and F = 42.0, *p* < 0.01, respectively) and by extract dose (F = 93.7, *p* < 0.01 and F = 30.1, *p* < 0.01). Globally, the RAEs of both parameters for this target species decreased with the higher extract dose of 5% (Figure 2). Both parameters were lowest when the *E. crus-galli* seedlings were exposed to *A. marina* aqueous extracts, with an average RAE of −68% and −55%, followed by *S. caseolaris* (−52% and −39%) and *A. corniculatum* (−14% and −19% for extract doses of 2.5% and 5%, respectively). For all the other tested source species, the hypocotyl length and biomass RAEs of this target species ranged from −13% to +4% and from −13% to +3%, respectively.

Source species and extract dose interacted in their effects on root growth for both target species (Table 1). *E. crus-galli* seedlings’ root growth was negatively affected in all the treatments with RAEs, ranging particularly low from −99% to −82%, when watered with *S. caseolaris* (2.5 and 5%), followed by *A. marina* (5%), *B. gymnorhiza* (5%), *A. marina* (2.5%) and *L. racemosa* (5%). When watered with *B. gymnorhiza* (2.5%), *A. corniculatum* (5%) and *S. apetala* (5%), the RAEs ranged from –64% to −47%.

In the case of *O. sativa* seedlings, highly inhibitory effects on root growth were also observed when exposed to *S. caseolaris* extracts at both 5% (RAE = −92%) and 2.5% (RAE = −81%). However, in contrast with *E. crus-galli*, *O. sativa* seedlings’ root growth was positively affected by certain treatments. The RAEs reached 65% and 47% when watered with *A. corniculatum* (2.5% and 5%, respectively) and 53% with *S. apetala* aqueous extracts (2.5%). This was the only assessed parameter that showed positive allelopathic effects above 15%. Still, in both cases, when the extract dose was increased, this stimulatory effect was diminished by 19% for *A. corniculatum* extracts and by 64% for *S. apetala*, with an inhibitory effect occurring for this source species extract at 5% (Figure 2).

*O. sativa* seedlings’ hypocotyl growth and biomass were also affected by the source species and extract dose in the interaction (F = 6.9, *p* < 0.01; F = 2.2, *p* < 0.05; and F = 8.3, *p* < 0.01, respectively). The only positive allelopathic effects observed for these two parameters on *O. sativa* were found for hypocotyl length when watered with *A. corniculatum* (2.5%) and for total biomass when watered with *S. apetala*, also at a lower dose (Figure 2). The hypocotyl length and biomass RAEs were equivalently reduced to below −50% when *O. sativa* was exposed to *A. marina* (5%), followed by *L. racemosa* (5%) aqueous extracts, and below −60% when exposed to *S. caseolaris* extracts, with an enhancement in inhibition observed at higher dose for this source species of 9% on average. These allelopathic effects of the source species depended on the dose in several cases. All the tested extracts induced stronger hypocotyl growth inhibition at a higher dose, except for *K. obovata*, and all the tested extracts induced higher biomass inhibition at a higher dose for *O. sativa*, except for *A. corniculatum*.

Finally, for both target species, the interaction of the tested extracts’ source species and dose had a significant effect on their numbers of roots. As assessed for hypocotyl length and biomass, the allelopathic effects on the *O. sativa* seedlings’ number of roots were found to be below −60% when watered with *A. marina* and *S. caseolaris* extracts, but at higher concentrations only. Root number RAEs ranging from −50% to −30% were observed for *O. sativa* seedlings in the presence of *B. gymnorhiza* (5%) and *S. caseolaris* (2.5%) aqueous extracts, and from −30% to −20% for *A. marina* (2.5%), *K. obovata* (5%) and *L. racemosa* (5%) (from lowest to highest). In the case of *E. crus-galli* seedlings, root number was also strongly affected when exposed to *S. caseolaris* (2.5% and 5%) and *A. marina* (5%) extracts with RAEs below −60% and found to range from −50% to −30% when exposed to *A. marina* (2.5%) and *B. gymnorhiza* (5%) (from lowest to highest). Only the latter three source species showed significantly lower RAEs at a higher dose for *E. crus-galli*, whereas for *O. sativa* seedlings, all the tested extracts induced stronger hypocotyl growth inhibition at the highest dose, except for *R. stylosa* and *S. apetala*.

#### 2.2.3. Selection of Candidate Species for Further Chemical Investigation

Based on these results, and with both target species belonging to the Poaceae family, we chose germination rate and root growth as the main parameters for the selection of candidate species producing allelopathic compounds. We also chose to set a threshold for RAE indices of −20% and 20% for negative and positive allelopathic effects, respectively, to select the highest allelopathic potentials that could occur on the field. According to these criteria, the source species *A. corniculatum* and *S. apetala* were selected as candidates and their chemical diversity was therefore investigated.

### 2.3. A. corniculatum and S. apetala Biomarker Selection and Annotation

*A. corniculatum* and *S. apetala* leaf extracts at a 2.5% dose both strongly stimulated *O. sativa* seedlings’ root growth while inhibiting *E. crus-galli*’s; these two aqueous extracts also led to the inhibition of *E. crus-galli*’s germination rate, while no effect was observed for *O. sativa* seedlings. For these reasons, the two species were selected for further phytochemical investigation, and their biomarkers and major compounds from their corresponding base peak chromatograms (BPC) were annotated. The metabolite fingerprints of the leaf methanolic extracts were investigated via liquid chromatography coupled with a high-resolution mass spectrometer (UHPLC-MS) or tandem mass spectrometry (UHPLC-MS/MS). When highlighting the fifty most discriminant biomarkers within all the tested mangrove species (Figure 1b), four were found to be specific to *A. corniculatum* and three to *S. apetala* (Table 2). The BPCs allowed us to point out four major compounds for *A. corniculatum* and six for *S. apetala* (Table 3, Appendix A). Their structures were putatively annotated by comparing their MS data with constructor (Bruker Daltonics^®^, Wissembourg, France), in-house (chemical ecology platform, MALLABAR, Marseille, France) and online databases, as well as the literature. The detailed annotation and MS/MS spectrum of each compound, with some of their chemical structures, are shown in the Appendix A.

Biomarker M781T518 (Appendix A), specific to *A. corniculatum* leaves, could correspond to a triterpenoid saponin derivative. Biomarker M943T476 (Appendix A) could be assigned to (3β,16α,20α)-3,16,28-trihydroxyolean-12-en-29-oic acid 3-{O-β-D-glucopyranosyl-(1→2)-O-[β-D-glucopyranosyl-(1→4)]-α-L-arabinopyranoside}, a saponin already described in leaves of *A. corniculatum* collected in Vietnam [41]. Biomarker M410T546 (Appendix A) could be a monoterpene sulfate. The BPC major compound Ac M1 (Appendix A) might be myricetin-3-glucoronid (Appendix A), found in the leaves of mangrove *Conocarpus erectus* [42]. The BPC major compound Ac M2 (Appendix A) could be a quercetin glucuronide already described in *Laguncularia racemosa* [43], while the BPC major compound Ac M3 (Appendix A) could be kaempferol-3-glucoside, also known as astragalin (Appendix A). Vinh et al., 2019 reported this metabolite in *A. corniculatum* leaves, supporting our assumptions [41]. Among the *A. corniculatum* biomarker and compounds, two remained unidentified.

In contrast with *A. corniculatum*, all the highlighted biomarkers specific to *S. apetala* corresponded to the main peaks of the BPC. Biomarker M481T560 (also the BPC major compound Sa M6) (Appendix A) of *S. apetala* was putatively annotated as 3,3′,4′-tri-O-methyl-ellagic acid 4-sulfate (Appendix A), which has recently been reported for the first time in mangrove species in the roots of *Lumnitzera littorea* and *L. racemosa* from Indonesia [40]. According to the same study, compound Sa M5 (Appendix A) could be assigned to a dimethyl-ellagic acid sulfate (Appendix A). Compound M607T418 (also compound Sa M4) (Appendix A) could be assigned to diosmin (Appendix A), a disaccharide derivative consisting of diosmetin substituted by a rutinose at position 7 via a glycosidic linkage. This assumption was further reinforced by the presence of diosmetin, reported in two species from the *Sonneratia* genus: *S. paracaseolaris*’ aerial parts [44] and *S. alba*’s leaves [45]. Biomarker M895T343 (also compound Sa M1) (Appendix A) could correspond to luteolin or kaempferol. Characteristic fragment ions observed in the MS/MS spectrum [46] and luteolin, already described in *S. apetala* fruits [47], allowed us to suggest that it is a luteolin glucoside. Although specifically more abundant in *S. apetala*, biomarker M895T343 was also detected in *S. caseolaris* and *B. gymnorhiza* extracts (Figure 1b). For these additional reasons, this compound could be a luteolin 7-O-β-glucoside already described in *S. caseolaris* leaves and fruits [48,49]. Compound Sa M3 (Appendix A) might be vitexin (Appendix A), a flavone glycoside already described in *S. apetala* leaves and branches collected in China [50]. Finally, compound Sa M2 (Appendix A) might be a 1,3,4,6-tetra-O-galloyl-β-D-glucose previously reported in the mangrove species *Excoecaria agallocha* [51].

**Table 2 plants-11-02464-t002:** List of *Aegiceras corniculatum* and *Sonneratia apetala* leaf methanolic extracts’ biomarkers detected via UHPLC-(-)-ESI-qToF.

	Biomarker	[M-H]− m/z	Adduct	RT (min)	Molecular Formula	Mass Error (ppm)	Mσ ^a^	MS/MS Fragment Ions (Relative Abundance in %)	Putative Identification	Reference
** *Aegiceras corniculatum* **	M943T476	943.4872	979.4685 [M+Cl]^−^ 1011.4771 [M-H+HCOONa]^−^	7.94	C_47_H_76_O_19_	3.8	36.7	943.4872 [M-H]^−^ (38.4), 781.4355 [M-H-Hexose]^−^ (100), 619.3829 [M-H-2Hexose]^−^ (63.2), 487.3426 [M-H-2Hexose-Pentose]^−^ [C_30_H_47_O_5_]^−^ (38.5), 275.0768 (42.2), 113.0243 (28.7), 101.0239 (33.8), 89.0238 (29.3)	(3β, 16α, 20α)-3,16,28-trihydroxyolean-12-en-29-oic acid 3-{O-β-D-glucopyranosyl (1→2)-O-[β-D-glucopyranosyl (1→4)]-α-L-arabinopyranoside}	[41]
	M457T495	457.1700	479.1545 [M-2H+Na]^−^ 523.1408 [M-H+HCOONa]^−^	8.25	C_21_H_30_O_11_ (or C_20_H_31_N_2_O_8_P)	−0.3 4.9	30.6 8.9	238.0854 (100.0), 235.1335 (2.7), 235.0973 (4.8), 220.0748 (5.3), 194.0945 (11.3), 192.0783 (3.1), 177.0913 (3.0), 165.0553 (5.5), 151.0393 (3.3)	Unknown	
	M781T518	781.4384	817.4162 [M+Cl]^−^ 849.4374 [M-H+HCOONa]^−^	8.64	C_41_H_66_O_14_	0.8	7.0	781.4384 [M-H]^−^ (31.7), 619.3849 [M-H-Hexose]^−^ (100), 601.3749 (22.7), 487.3456 [M-H-Hexose-Pentose]^−^ [C_30_H_47_O_5_]^−^ (10.3), 101.0242 (19.3), 89.0244 (18.7)	Triterpenoid saponin	[41]
	M410T546	409.1175	477.1044 [M-H+HCOONa]^−^	9.10	C_16_H_26_O_10_S	−2.3	9.5	241.0029 [C_6_H_9_O_8_S]^−^ (9.1), 138.9708 [C_2_H_3_O_5_S]^−^ (4.4), 96.9600 [HSO_4_]^−^ (100)	Monoterpene sulfate	
** *Sonneratia apetala* **	M895T343 Sa M1	447.0941	505.0515 [M+NaCl-H]^−^ 515.0801 [M-H+HCOONa]^−^	5.72	C_21_H_20_O_11_	−1.8	16.5	357.0624 (22.9), 339.0514 (21.3), 327.0516 (79.4), 311.0565 (26), 298.0483 (100), 285.0405 [M-H-Hexose]^−^ [C_15_H_9_O_6_]^−^ (51.4), 284.033 [M-H-Hexose]^−^ (38.1), 199.0407 (5.4), 175.0405 (8.6), 151.0039 (3.4), 133.0298 (36.7)	Luteolin 7-O-β-glucoside	[46,47,48,49]
	M607T418 Sa M4	607.1671	643.1442 [M+Cl]^−^ 675.1545 [M+HCOONa-H]^−^	6.97	C_28_H_32_O_15_	−0.2	5.0	299.0563 [M-H-rutinose]^−^ (53.5), 284.0329 [M-H-rutinose-CH_3_]^−^ (100)	Diosmin ^a^	[44,45]
	M481T560 Sa M6	423.0023	480.9607 [M+NaCl-H]^−^ 490.9817 [M+HCOONa-H]^−^	9.34	C_17_H_12_O_11_S	0.2	10.8	343.0471 [M-H-SO_3_]^−^ (0.3), 328.0227 [M-H-SO_3_-CH_3_]^−^ (20.6), 312.9992 [M-H-SO_3_-2CH_3_]^−^ (100), 297.9757 [M-H-SO_3_-3CH_3_]^−^ (51.4), 285.0042 (17.2), 269.9807 (8.8)	3,3′,4′-trimethylellagic acid 4-sulfate	[40]

^a^ Confirmed using commercial standards (in-house MS/MS spectra database).

**Table 3 plants-11-02464-t003:** List of *Aegiceras corniculatum* and *Sonneratia apetala* leaf methanolic extracts’ base peak chromatogram (BPC) major compounds detected via UHPLC-(-)-ESI-qToF.

	Compound	[M-H]^−^ *m*/*z*	Adduct	RT (min)	Molecular Formula	Mass Error (ppm)	Mσ ^a^	MS/MS *m*/*z* Fragment Ions (Relative Abundance in %)	Putative Identification	Reference
** *Aegiceras corniculatum* **	Ac M1	493.0607	515.0438 [M+Na-2H]^−^	5.75	C_21_H_18_O_14_	3.3	23.1	317.0291 [M-H-glucuronic acid]^−^ (62.3), 271.0237 (29.1), 261.0395 (12.5), 243.0289 (11.0), 178.9980 (56.8), 163.0030 (11.4), 151.0031 (100), 137.0239 (53.2), 109.0291 (12.9), 107.0134 (17.6)	Myricetin-3-glucuronide	[42,43,52]
	Ac M2	477.067	499.0502 [M+Na-2H]^−^	6.37	C_21_H_18_O_13_	1.0	21.4	301.0350 [M-H-glucuronic acid]^−^ (72.3), 283.0242 (17.0), 255.0295 (26), 245.0451 (27.3), 178.9983 (32.4), 163.0033 (20.7), 151.0034 (100), 121.0293 (30.3), 109.0292 (19.7), 107.0136 (23.2)	Quercetin glucuronide	[43]
	Ac M3	447.0924	483.0692 [M+Cl]^−^ 515.0804 [M+HCOONa-H]^−^	6.90	C_21_H_20_O_11_	2.0	15.2	300.0273 (6.6), 284.0323 (38.7), 271.0242 (10.0), 255.0296 (100), 227.0347 (85.0)	Kaempferol-3-O-glucoside ^a^ (Astragalin)	[41]
	Ac M4	451.1638	519.1499 [M+HCOONa-H]^−^	7.08	C_19_H_32_O_10_S	1.2	22.5	451.1636 [C_19_H_31_O_10_S]^−^ (25.8), 256.9969 [C_6_H_9_O_9_S]^−^ (6.3), 241.0015 [C_6_H_9_O_8_S]^−^ (6.5), 177.0401 [C_6_H_9_O_6_]^−^ (2.8), 138.9704 [C_2_H_3_O_5_S]^−^ (2.2), 96.9597 [HSO_4_]^−^ (100), 79.9571 [SO_3_]^−^ (3.5)	Unknown	
** *Sonneratia apetala* **	Sa M1									
	Sa M2	787.1008	809.0818 [M+Na-2H]^−^ 823.0780 [M+Cl]^−^	6.00	C_34_H_27_O_22_	−1.1	9.4	635.0915 (0.3), 483.0789 (1.1), 465.0678 (50.2), 313.0569 (47.7), 295.0463 (18.8), 169.0145 (100), 125.0247 (26.8)	Tetragalloyl glucose	[51,53,54]
	Sa M3	431.0998	499.0861 [M+HCOONa-H]^−^	6.25	C_21_H_20_O_10_	−3.4	9.6	323.0566 (13.1), 311.0569 (31.2), 295.0618 (11.3), 283.0617 (100), 281.0460 (20.6), 269.0457 (11.1), 117.0350 (21.6)	Vitexin ^a^	[50]
	Sa M4									
	Sa M5	408.9880	329.0310 [M-SO_3_-H]^−^ 476.9758 [M+HCOONa-H]^−^	7.82	C_16_H_10_O_11_S	−2.1	20.9	329.0303 [M-H-SO_3_]^−^ (0.6), 314.0067 [M-H-SO_3_-CH_3_]^−^ (37.0), 298.9834 [C_14_H_3_O_8_]^−^ [M-H-SO_3_-2CH_3_]^−^ (100), 270.9882 [C_13_H_3_O_7_]^−^ (48.9), 242.9934 [C_12_H_3_O_6_]^−^ (2.6)	Dimethyl ellagic acid sulfate	[40]
	Sa M6									

^a^ Constructor statistical match factor (*m*/*z* defect and comparison of experimental vs. theoretical isotopic patterns).

## 3. Discussion

### 3.1. Leaf Chemical Fingerprint Similarities According to Phylogenetic Proximity

The first aim of our study was to compare the global metabolic contents of eight mangrove species’ leaves through an untargeted LC-MS-based metabolomics approach. Chemical fingerprinting comparison allowed us to show the global metabolic similarities according to phylogenetic proximity. Indeed, the three mangrove species belonging to the Rhizophoraceae family formed a specific cluster (with the *K. obovata* and *R. stylosa* samples overlapping on the two main components), while the species from the *Sonneratia* genus formed one as well. Nevertheless, when considering species groups according to their allelopathic effects, this chemical clustering did not allow us to show a clear relationship between allelopathy and chemotaxonomy.

### 3.2. Contrasted Allelopathic Effects Depending on Mangrove Species

The second aim of our study focused on the allelopathic potential of leaf macerates from these species and their effect on the development of plants from an adjacent agrosystem to the mangrove in the Red River Delta: *E. crus-galli*, the main weed of rice culture and rice itself. Both germination and early seedling growth are key stages in plant development, and are therefore potentially more sensitive to the effects of allelochemicals from neighboring plants [19]. The results of our study showed that leaf water macerates of mangrove species can affect the germination and growth of target species in different ways depending on their sensitivity to allelochemicals and extract dose. Indeed, the target species’ response to allelochemicals depends on the considered species [13].

Germination rate is a critical parameter for crop plant development as well as for weed settlement success [55]. Rice early root growth has previously been reported as being positively correlated with rice yield [56,57]. Additionally, previous studies have shown that root length is the best indicator of the allelopathic effects of plant extracts, especially in *Medicago sativa*, often used in bioassays. Indeed, the growth of its roots is more sensitive to phytotoxic compounds than the germination of its seeds or the growth of the hypocotyl of its plantlets [58]. For these reasons, we chose to focus on these two parameters to classify the assessed mangrove source species according to their effects on both target species. In order to select the relevant source species for further field bioassays, we also chose to focus on marked allelopathic effects (RAE < −20 for inhibiting effects; RAE > 20% for stimulating effects), since effects observed in vitro are generally diminished in field conditions [17,59].

#### 3.2.1. Allelopathic Effects on Germination

Concerning the effects on the germination of *E. crus-galli*, our results showed that four source species inhibited its germination rate: *B. gymnorhiza*, *A. marina*, *S. apetala* and *A. corniculatum*. This inhibitory effect is more important at a high dose. In the case of *O. sativa*, *B. gymnorhiza* is the only source species that reduced its germination rate at any dose. These results match with those of Chen and Peng [30], who have previously shown that specialized metabolites contained in leaf aqueous extracts of *A. marina* and *B. gymnorhiza* decreased the germination rates of model species used in bioassays, *Lactuca sativa* L. and *Raphanus sativus* L.

#### 3.2.2. Allelopathic Effects on Root Growth

If we consider the root growth of *E. crus-galli*, almost all the source species had an inhibitory effect. This effect increases with the dose of extract, except in *S. caseolaris*. Similarly, Xuan et al. [60] showed that extracts of neem (*Azadirachta indica*. A. Juss) bark and leaves inhibited the root elongation of *E. crus-galli*, highlighting the susceptibility of this weed.

Considering the root growth of *O. sativa*, our results showed four groups of source species according to their different effects on this parameter. The first group consists only of *S. caseolaris*, which strongly inhibited rice root growth at all doses, with a greater inhibitory effect at high doses. Our results are in accordance with those of Hasegawa et al. [61] who showed that *S. caseolaris* protoplasts exhibited high inhibitory effects on lettuce protoplast cell division in coculture. Indeed, allelochemicals can alter the enzymatic activity, as well as the physiological processes, of cell division and elongation [15] and/or increase the permeability of the plasma membrane of cells; thus, they can limit the absorption of nutrients by the root [62].

The second group consists of source species that show an inhibitory effect on rice root growth at high doses but whose effect disappears at low doses: *A. marina*, *B. gymnorhiza* and *K. obovata*. Similarly, Chen and Peng [30] showed that leaf aqueous extracts of *A. marina* and *B. gymnorhiza* inhibited the growth of cabbage (a model species) and *K. candel* (species following *A. marina* in mangrove succession). In addition, Lang et al. [63] demonstrated the allelopathic potential of *K. obovata* against one of its associated species *A. corniculatum* in a Chinese mangrove, indicating that allelopathy plays a role in mangrove ecosystems.

The third group consists of *S. apetala* and *L. racemosa*, which exhibited both inhibiting and stimulating effects with high and low doses, respectively, with a more important stimulating effect from *S. apetala*. Liu et al. [64] described the dose–response relationship as, usually, an inverted U-shape in the science of allelopathy. Similarly to our results, An et al. [65] described the stimulation–inhibition phenomenon in allelopathy mathematically and defined this type of response as a biological property of allelochemicals. Considering the source species, a previous study showed the allelopathic effect of *S. apetala* on the development of the invasive species *Spartina alterniflora* Loisel (Poaceae) [32]. However, to our knowledge, *L. racemosa*’s allelopathic potential is shown for the first time in the present work.

Finally, the last group is the most interesting since it consists of the species that inhibits the root growth of *E. crus-galli* while stimulating that of *O. sativa*, regardless of the extract dose in both cases. This species is *A. corniculatum*, which is already cited in previous studies; for example, Liu et al. [66] showed that gallic acid isolated from root exudates of *A. corniculatum* exhibited an inhibiting effect on the algae *Cyclotella caspia*.

#### 3.2.3. *A. corniculatum* and *S. apetala* Are Interesting Candidates for Further Investigations

From the consideration of the combined results on the root growth of *E. crus galli* and *O. sativa*, we distinguished two candidate species to be used in the control of the weeds *E. crus-galli*, *S. apetala* and *A. corniculatum*. Still, according to our results, the contrasted effects of *S. apetala* extracts on *O. sativa* root growth at high and low doses indicate that special attention should be given to this parameter in further investigations. For *A. corniculatum*, however, this contrast between doses did not occur. Further investigations in the field are necessary.

### 3.3. A. corniculatum and S. apetala Chemical Investigation

The third aim of this study was to highlight the assessed species-specific biomarkers of mangroves showing interesting allelopathic activities as potential bioherbicides against barnyard grass. Here, a total of nine phenolic compounds were annotated in *A. corniculatum* and *S. apetala* leaf methanolic extracts. Three flavonoids (flavonol glycosides) were associated with *A. corniculatum*, while three flavonoids (flavone glycosides), a tannin and two sulfated ellagic acid derivatives were described in *S. apetala*. Moreover, two saponins were found specifically in *A. corniculatum*. Even if these metabolites were highlighted in methanol extracts, all the annotated compounds’ structures indicate their water solubility; we may therefore suggest their presence in the assessed water macerates.

#### 3.3.1. *A. corniculatum* Biomarkers and Compounds as Potential Allelochemicals

*Aegiceras corniculatum* has been phytochemically investigated in several previous studies [37,67,68] and pharmacological assessments showed antimicrobial, anti-inflammatory, antioxidant and cytotoxic activities in this plant [41,69,70].

Several saponins, triterpenes, alkaloids, sterols, hydroquinones and flavonoids have been previously reported in *A. corniculatum* [68,69]. Our results highlighted two triterpene saponin biomarkers found in *A. corniculatum*; one of them (M943T476) might be (3β, 16α, 20α)-3,16,28-trihydroxyolean-12-en-29-oic acid 3-{O-β-D-glucopyranosyl (1→2)-O-[β-D-glucopyranosyl (1→4)]-α-L-arabinopyranoside}, already described by Vinh et al. [41] in *A. corniculatum* leaves. Saponins are described as a potential source of bioherbicides, and they have been most widely studied for allelopathic interactions with model species *M. sativa* [71]. Their effect can be either inhibitory or stimulatory to seed germination and the development of root and shoot. Ghimire et al. [72] showed the autotoxic effect of saponins isolated from *M. sativa*, where treated seedlings experienced a lack of root hair, reduced seedling taproot and complete root color loss. On the same model, Oleszek et al. [73] suggested that these effects are dependent on saponin structures and concentrations. Reigosa et al. [71] also reported these compounds as being toxic to barnyard grass and cheat (*Bromus secalinus*).

Among the three flavonol glycosides identified in *A. corniculatum*, myricetin-3-glucuronide (Ac M1) and quercetin glucuronide (Ac M2) were described for the first time. Astragalin (Ac M3) has already been reported in the leaves of this plant [41]. This compound has previously shown allelopathic potentialities against several crop model species, where Ladhari et al. [74] observed a negative correlation between astragalin content in *Ficus carica* extracts and target species’ root and shoot growth. Their study revealed that the cultivars of *F. carica* with potent allelopathic potential are provided by the highest level of astragalin for the cultivar Bouhouli. This compound may therefore also contribute to the *A. corniculatum* herbicidal activities observed in our study.

#### 3.3.2. *S. apetala* Biomarkers and Compounds as Potential Allelochemicals

*Sonneratia apetala* phytochemicals were also described [47,75,76], and studies have shown antimicrobial, antioxidant and anti-inflammatory activities in this species [50,77].

Studies report the presence of polysaccharides, alkaloids, carbohydrates, saponins, tannins and flavonoids in *S. apetala* mangroves [78,79]. In the present study, all the annotated metabolites in *S. apetala* leaves’ methanolic extracts were reported for the first time, except for vitexin (Sa M3), which was found to be a major constituent (along with gallic acid and isorhamnetin) in the branches and leaves of *S. apetala* collected in China [50]. A recent study allowed researchers to identify this flavone glucoside as an allelochemical with natural herbicide activities against *Phalaris minor*, *Avena fatua*, *Chenopodium album* and *Rumex dentatus* growth through the bio-guided fractionation of *Lantana camara* leaves’ extracts [80]. Reversely, Kalinova et al. [81] tested the effect of vitexin on eight plant species and showed no effect or a stimulative effect on their growth. *E. crus-galli* growth was not affected by vitexin in their study, but stimulating effects on *Trifolium repens*, *L. sativa* and *Sinapis alba* growth were shown. Still, these results were obtained using bioassays conducted in the dark, while Basile et al. [82] observed vitexin inhibition of seed germination in *R. sativus* when sown in the light after 24 and 48h, and no effect when kept in the dark. The bioassays of the present study were conducted in the light; vitexin may therefore be considered as potentially involved in the observed *E. crus-galli* seed germination inhibition of *S. apetala* aqueous extracts. Moreover, the contrasted effect of this compound on different target species’ growth is in accordance with our findings with *S. apetala* aqueous extracts since *O. sativa* germination was not affected.

#### 3.3.3. First Description of Sulfated Phenolics in *S. apetala*

To our knowledge, sulfated ellagic acids are described for the first time in *S. apetala* leaves and in the *Sonneratia* genus in general in this study. Manurung et al. [40] reported these compounds in the roots of *Lumnitzera littorea* and *L. racemosa* mangroves with several other sulfated phenolics and suggested that their accumulation could be organ-specific. Here, we reported the putatively annotated 3,3′,4′-tri-O-methyl-ellagic acid 4-sulfate (M481T560, Sa M6) in the leaves of *S. apetala* and *L. racemosa* for the first time, highlighting their presence in the aerial parts as well.

Sulfated phenolics are naturally occurring in plants and seagrass [83]. Among them, sulfated ellagic derivatives remain scarcely described. Owczarek et al. [84] reported Sa M6 in the rhizomes of *Geum rivale* L. (Rosaceae), and Terashima et al. [85] reported it in the roots of *Potentilla candidans* Humb. & Bonpl. Ex Nestl (Rosaceae). Dimethyl-ellagic acid sulfate (Sa M5) also found to be a BPC major constituent of *S. apetala* leaves in our study, was found in *Myrciaria cauliflora* (Mart.) O. Berg (Myrtaceae) [86].

Sulfation is known to increase phenolics’ water solubility while decreasing their toxicity [83]. It is often encountered in aquatic plants, suggesting it as an ecological trait, and is potentially involved in adaptation to salty soils. The high number of sulfates found in mangrove sediments [87] probably facilitates the production of sulfated derivatives by mangrove species. To our knowledge, *L. littorea*, *L. racemosa* trees and fern *Acrostichum aureum* are the only mangrove species in which sulfated specialized metabolites have been reported [40,88]. These results suggest that there might be more sulfated phenolic compounds in the assessed source species to be investigated.

In terms of activity, the compound Sa M6 might be involved in the common observed allelopathic effects of *S. apetala* and *L. racemosa* aqueous extracts on *O. sativa* root growth since they similarly induced a reduction at high concentrations and an increase in root lengths at lower concentrations in this study. In addition, sulfated flavonoids were reported to regulate plant growth by influencing the efflux inhibition of auxins, which are involved in root development [83]. We hypothesize that sulfated ellagic acids may play a similar role in the effects of *S. apetala* extracts on the root growth regulation of rice in our bioassay. The isolated compounds’ effects need to be further assessed.

### 3.4. Perspectives for Further Field Assays

Further studies should be conducted in situ. Lee et al. [89] applied a mixture of rice bran and pine needles to test the inhibitory effects on rice weeds and to increase rice yield. The weed control efficacy of the combined application with pine leaves was 100% 70 days after transplanting the rice and was maintained 90 days after transplanting. A similar use of mangrove leaves could be considered and tested to provide an environmentally friendly weed management technique in rice fields. Indeed, this practice would use natural compounds that are more easily degradable [15] and resources from an ecosystem close to the rice fields. Our findings show that owing to their contrasted allelopathic effects on barnyard grass and rice, and to the presence of metabolites that have previously been reported as allelochemicals, *A. corniculatum* and *S. apetala* leaves are promising candidates as alternatives to synthetic herbicide.

## 4. Materials and Methods

### 4.1. Sampling Site

The sampling study site was located at the Mangrove Ecosystem Research Station (MERS) in a mangrove plantation in the Giao Lac Commune, Giao Thuy District, Nam Dinh Province in North Vietnam (20°1427.2″ N 106°30′52.2″ E). This region has a tropical climate with an annual average temperature of 23 °C and an annual average of precipitations of 1800 mm. The maximum and minimum monthly average temperatures are around 28 °C in July and 16 °C in January, respectively [90]. Giao Lac is a rural commune where fishery and agriculture are developed. Among the 600 ha of intertidal zone, 345 ha is covered by mangroves.

### 4.2. Material Collection

Leaves from eight species of mangroves were collected:-Three species belonging to the Rhizophoraceae family: *Rhizophora stylosa* Griff*, Bruguiera gymnorhiza* Buch ham and *Kandelia obovata* (L.) Lam.-Two from the *Sonneratia* genus (Lythraceae): *S. caseolaris* (L.) Engl. and *S. apetala* Sheue, Liu & Yong sp.nov-Three other species belonging to different families: *Aegiceras corniculatum* (L.) Blanco (Primulaceae), *Lumnitzera racemosa* Willd (Combretaceae) and *Avicennia marina* (Forssk.) Vierh (Acanthaceae).

The species were identified according to the plant morphology description [91,92]. After sampling, the leaves were freeze-dried and kept in the dark pending bioassays and chemical analysis. The seeds of *Oryza sativa* L. and *Echinochloa crus-galli* L. (Poaceae) came from the French Rice Center and the Caen Botanical Garden. In order to eliminate dormancy, the seeds of *E. crus-galli* were soaked in 95% sulfuric acid for 15 min [93]. They were abundantly washed with deionized water before their use in bioassays.

### 4.3. Allelopathy Bioassay

Laboratory bioassays are often the first step of allelopathic studies in the understanding of eco- or agrosystems’ functioning. Even if the conditions are not the same as in natura, they are useful to point out the activity of allelochemicals [94]. Water-soluble compounds are probably the most involved in allelopathy [95]. For these reasons, foliar leachates were used for the bioassays. The stock solution of the leaves’ extracts was prepared by soaking 5 g (dry weight) in 100 mL of deionized water (5%) for 24 h in darkness. A diluted solution (2.5%) was prepared from the stock solution. Extracts were stored at 4 °C for a maximum of 7 days before starting the experiment.

The bioassays were conducted in plastic culture boxes (100 × 80 × 80 cm) with 15.0 g (±0.1 g) of perlite to allow for root system development. In each box, 20 seeds of each target species were sown and watered with 90 mL of deionized water (control) or with leaf aqueous extract (2.5% or 5.0%). Five replicates were performed for each treatment (species × extract dose). The bioassays were conducted under a 14 h photoperiod (PAR = 110 µmoL·m^−1^·s) and a controlled temperature (28 ± 1 °C during the day and 25 ± 1 °C during the night) until 10 days after germination.

### 4.4. Germination Parameters

We studied two sets of plant traits reflecting germination and seedling growth. The relative germination was calculated using the following formula:(1)Germination (%)=NGNtotal×100
where *N_G_* is the number of germinated seeds in a treatment and *N_total_* is the total number of seeds in one box. The germination velocity was calculated using the Kotowski velocity coefficient:(2)Cv=∑ Ni∑ Ni Ti×100
where *N_i_* is the number of seeds germinated at time *I*, and *T_i_* is the number of days since the start of the experiment.

### 4.5. Growth Parameters

To estimate seedling growth, the lengths of the root and shoot were measured for each individual 10 d after germination (accuracy: 1 mm). Seedling biomass was evaluated by weighting each individual after oven-drying for 24 h at 60.0 °C. The number of adventive roots was observed as it is an important parameter for Poaceae species, which develop root systems characterized by a high degree of root initiation from the belowground basal nodes of the shoot [96]. For all the parameters, we calculated the Relative Allelopathic Effect (RAE) using the following formula:(3)RAE=O−CC×100
where *O* is the observed value when a target species is exposed to allelopathic compounds and *C* is the mean value measured for the control [20].

### 4.6. Metabolomic Analysis, Data Pre-Processing and Annotation of Metabolites

Leaves from six individuals of the eight species were frozen, freeze-dried and ground. A mass of 200 mg was added to 400 mL of methanol. The mixture was placed in an ultrasonic bath for 5 min at room temperature and filtered (Restek, PTFE 13 mm, 0.22 μm, Agilent Technologies^®^, Les Ulis, France). A total of 4 µL was injected into a chromatographic instrument equipped with an RS Pump, autosampler and thermostated column compartment, and a UV diode array detector (Dionex Ultimate 3000, Thermo Scientific, Illkirch-Graffenstaden) coupled with a high-resolution mass spectrometer (HRMS) equipped with an electrospray source (ESI) and a quadrupole time-of-flight analyzer (QTOF) (Impact II, Bruker Daltonics^®^, Fällanden). Analytical blanks and a pool solution consisting of 20 µL of each sample were also prepared, and all the samples were randomly injected to avoid analytical variability as previously described by Paix et al. 2019 and Favre et al. 2017 [35,36]. The elution rate was set to 0.5 mL·min^−1^ at a constant temperature of 40 °C. The chromatographic solvents were composed of A: H_2_O and B: ACN each containing 0.1% formic acid (*v*/*v*). Separation was performed using an Acclaim RSLC C18 column (150 mm × 2.1 mm, 2.2 µm, Thermo Scientific^®^, Illkirch-Graffenstaden). The elution gradient consisted of 0–2 min at 10% B; 2–9 min at 10–50% B; 9–11 min at 50% B; 11–13 min at 100% B; and 13–15 min at 10% B for a total run of 15 min. Mass spectra were acquired in negative mode from 50 to 1200 amu. A formate/acetate solution forming clusters was injected before the sequence to calibrate the mass spectrometer. The solution was automatically injected before each sample for internal mass calibration. The capillary voltage was set to 3000 V with N_2_ as a nebulizing gas at a pressure of 3.5 bar, for 12 L·min^−1^, at a temperature of 200 °C. Collision-induced dissociation (CID) with a collision energy ranging from 40 to 70 eV allowed tandem mass spectra acquisition. The raw UHPLC-HRMS data were analyzed and converted into netCDF files using Data Analysis (version 5.0, Bruker Daltonics^®^). These files were then preprocessed (via chromatographic peak detection, integration, alignment and grouping) using the open-source XCMS package (version 3.6.2; [97]) in the R 3.6.0 version environment. After statistical analysis, the most discriminant features were selected for putative annotation.

A total of 12 compounds were putatively identified using a dereplication strategy [98,99]. Briefly, using the Data Analysis software (version 5.0, Bruker Daltonics^®^), chromatographic and spectral datasets were processed, and molecular formulae were deduced from the MS spectra of interest. With a 5 ppm error threshold for mass accuracy and the isotopic pattern quality mSigma (Mσ) kept low, C, H, S and O were selected for elemental composition to reduce the potential candidates. Regular detection of the adduct ions [M+HCOONa-H]^−^, [M+Cl]^−^ and [M+NaCl-H]^−^ allowed us to better support the identification of the pseudo-molecular ion [M-H]^−^. For each deduced raw formula, the KNapSAcK, the Dictionary of Natural Products, Sci Finder and PubChem public databases were interrogated to report known metabolites and, when necessary, eliminate formulae that were not referenced [34]. Corresponding HR-MS/MS spectra comparison with an in-house database built using commercial standards (MALLABAR chemical ecology platform, Marseille, France) and a constructor commercial database (Bruker Sumner MetaboBASE Plant Library) was carried out.

### 4.7. Statistical Analyses

The differences in RAEs on plant growth (root, hypocotyl and biomass) and germination (rate and velocity) according to the source species (*A. corniculatum*, *A. marina*, *B. gymnorhiza*, *K. obovata*, *L. racemosa*, *R. stylosa*, *S. apetala* and *S. caseolaris*), extract dose (2.5 and 5%) and their interaction were tested using two-way ANOVAs followed by a post hoc Tukey test for pairwise comparison. The normality and homoscedasticity of the model residuals were visually checked. A heatmap representation of the calculated RAEs was generated using the *pheatmap* package in R. The UHPLC-MS preprocessed data were log10-transformed and mean-centered. Principal component analysis (PCA) was performed with all 399 detected features and a hierarchical clustering heatmap (with a Euclidean distance measure and a Ward clustering method) was built with the 50 most discriminant features (one-way ANOVAs followed by a post hoc Tukey test) using the MetaboAnalyst 4.0 online resource (http://www.metaboanalyst.ca, accessed on 4 January 2022). The corresponding figures were exported from this platform. Permutational multivariate analysis of variance (PERMANOVA) analysis (*vegan* package) was then performed (10,000 permutations) to test the effect of species on all the variables. Apart from the MetaboAnalyst analyses, all the statistical tests and figures were performed and generated using R software (version 3.6.0, The R Foundation for Statistical Computing, Vienna, Austria).

## 5. Conclusions

This study showed global metabolite fingerprint similarities according to the phylogenetic proximity of the studied mangroves. Allelopathic activity assessments allowed us to classify the different species into several groups: those that were toxic for both *E. crus-galli* and *O. sativa* (*S. caseolaris*, *A. marina*, *L. racemosa*, *B. gymnorhiza* and *K. obovata*), those that did not show any notable effect for either of the target species (*R. stylosa*), and those that were toxic for *E. crus-galli* and neutral or stimulating for *O. sativa* (*A. corniculatum* and *S. apetala*). The species from this last group represent interesting candidates for the development of plant-based bioherbicides and for further investigations in situ. A total of 12 compounds were annotated in these species’ leaf methanolic extracts and could be allelochemicals. Three flavonol glycosides and two saponins were associated with *A. corniculatum*, while three flavone glycosides, a tannin and two sulfated ellagic acid derivatives were described in *S. apetala*. Allelopathic strategies may be a useful tool for working towards sustainable agriculture using bioherbicides as a crucial weed control element and as an alternative to synthetic herbicides.

## Figures and Tables

**Figure 1 plants-11-02464-f001:**
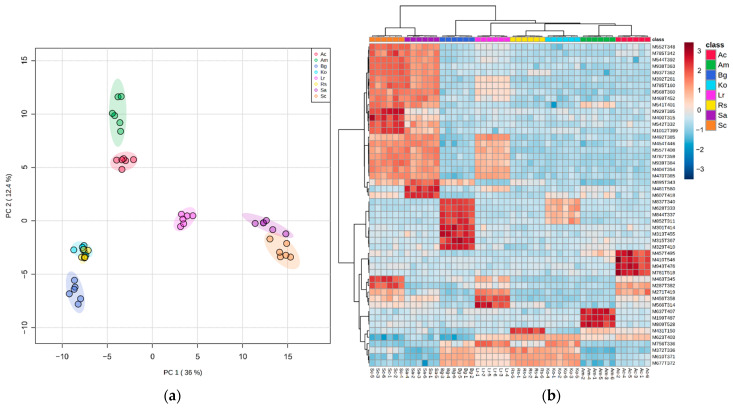
UHPLC-MS fingerprints of leaf methanolic extracts of *Aegiceras corniculatum* (red), *Avicennia marina* (green), *Lumnitzera racemosa* (pink), *Bruguiera gymnorhiza* (dark blue), *Kandelia obovata* (light blue), *Rhizophora stylosa* (yellow), *Sonneratia apetala* (purple) and *S. caseolaris* (orange): (**a**) principal component analysis where each sample is represented by a dot; (**b**) hierarchical clustering heatmap of most significant *m*/*z* features amongst species groups (ANOVA, *p* < 0.05). Each colored cell corresponds to a normalized, log10-transformed and mean-centered abundance value. Samples are in rows and *m*/*z* features in columns.

**Figure 2 plants-11-02464-f002:**
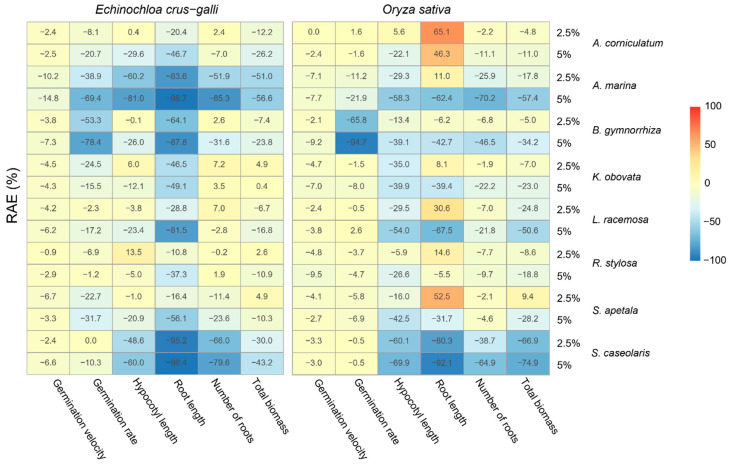
Heatmap representing Relative Allelopathic Effect (RAE) indices of 8 mangrove species’ (*Aegiceras corniculatum*, *Avicennia marina*, *Bruguiera gymnorhiza*, *Kandelia obovata*, *Lumnitzera racemosa*, *Rhizophora stylosa*, *Sonneratia apetala* and *Sonneratia caseolaris*) leaf aqueous extracts on germination (rate and velocity) and growth parameters (hypocotyl length, root length, number of roots and total biomass) of *Echinochloa crus-galli* and *Oryza sativa* seedlings 10 days after germination. The extracts were prepared at 2.5% and 5%.

**Table 1 plants-11-02464-t001:** Results of two-way ANOVAs testing for the effects of aqueous extracts’ source species (*Aegiceras corniculatum*, *Avicennia marina*, *Lumnitzera racemosa*, *Bruguiera gymnorhiza*, *Kandelia obovata*, *Rhizophora stylosa*, *Sonneratia apetala* and *S. caseolaris*), dose (2.5 and 5%) and their interactions on the Relative Allelopathic Effects (RAEs) on seed germination (*n* = 80 boxes) and root length (*n* = 908–1301 seedlings) of the two target species (*Echinochloa crus-galli* and *Oryza sativa*) 10 days after germination (Df: degrees of freedom; *p*-values are indicated with the respective symbols, * *p* < 0.05 and *** *p* < 0.001).

	RAE on Germination Rate (%)			RAE on Root Length (%)		
	Df	F-Value	*p*-Value	Df	F-Value	*p*-Value
*Echinochloa crus-galli*						
Source species (S)	7	13.3	***	7	201.8	***
Dose (D)	1	6.1	*	1	173.8	***
S × D	7	1.2		7	11.0	***
Residuals	64			892		
*Oryza sativa*						
Source species (S)	7	137.1	***	7	287.6	***
Dose (D)	1	13.3	***	1	301.6	***
S × D	7	4.7	***	7	31.6	***
Residuals	64			1285		

## Data Availability

Supplementary data are available on demand.

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
