# Peer review of "Allelopathic Potential of Mangroves from the Red River Estuary against the Rice Weed Echinochloa crus-galli and Variation in Their Leaf Metabolome"

_plants, 2022, doi:10.3390/plants11192464_

Round 1

Reviewer 1 Report

1. Please recheck the Latin name of Bruguiera gymnorrhiza” in The Plant List (Home — The Plant List).

2. Please supplement the reference of the formula for calculating RAE.

3. It is noted that the allelopathy bioassay employed an aqueous extract, whereas the metabolite analysis used a methanol solvent. How can the two results be reconciled?

4. In table 1, if “n=80” means the samples for test germination (the total plastic boxes), the residuals are incorrect, maybe it is “64”. Moreover, the position of “n=80” need coherence.

5. In table 1, What means about “n=908”? If it represents the total estimated population that survived, please add a note in the table or text in the Methods. However, it's not rigorous to use 908 samples, the survived individuals in one plastic box are measured repetition rather than actual repetition.

6. Is it correct that in figure 2, while certain values are positive, the color blocks that correspond to them appear blue?

7. In the Introduction, “To date, only a few studies reported the allelopathic potential of mangroves species in weed management [42,43].” However, In the Discussion, “About source species, a previous study showed the effect of S. apetala volatiles on the development of the invasive species Spartina alterniflora Loisel (Poaceae) [42]. To our knowledge, this is the first time that the potentialities of L. racemosa are highlighted.”Whether the results of the two literatures are consistent or different, what are the reasons for the differences? And how about references “[43]”?

8. Is it necessary to describe biomarkers selection and annotation in detail? Please describe the results briefly.

9. It would be better to divide content into several topics and set small headings to discuss. There are too many paragraphs in the Discussion, and the logical and hierarchical relationships between paragraphs are somewhat confusing.

10. The discussion focuses on the discussion of the results, which should not spend a lot of time repeating the introduction background content.

11. Please refine the language and delete the references that have no important correlation with this study. There are too many references.

Reviewer 2 Report

In this study, authors tried to compare the leaf chemical composition of eight mangrove species, to evaluate their allelopathic potential on the germination and growth of E. crus-galli, and check that the potential effects observed on E. crus-galli are not toxic for O. sativa. Results showed that A. corniculatum and S. apetala leaves are promising candidates as an alternative to synthetic herbicide. By UPLC-MS/MS analysis, phytochemical investigations of A. corniculatum and S. apetala indicated that both species showed putative flavonoid glucosides as BPC major compounds. This manuscript is well organized and comprehensively described. The research design is appropriate and the content is significantly. I recommend this manuscript to be accepted after minor revision.

Minor commends

A.     Abstract

1.     In the abstract, the name of identified compounds should be presented.

2.     Please change the submitting template from 2021 to 2022.

3.     In the new form, the number of lane should be remained.

B.     Results

1.     Please added the whole figure of UPLC or added the relative content of each identified compounds to prove the reality of concentration in the leaves.  

C.     Discussion

1.     In the Page 6, Lane 15. This sentence should be the next paragraph of the article.

D.     Materials and Methods

1.     Please described how to identified the mangroves species in this study. By molecular or phenotype methods.

2.     Please added the figures of LC-MS/MS patterns of each identified compounds in the part of supporting information.
